# Predicting knee adduction moment response to gait retraining with minimal clinical data

**Nataliya Rokhmanova**[1,2], **Katherine J. Kuchenbecker**[2], **Peter B. Shull**[3], **Reed Ferber**[4], **Eni Halilaj**[1]*

**1** Department of Mechanical Engineering, Carnegie Mellon University, Pittsburgh, Pennsylvania, United States of America, **2** Max Planck Institute for Intelligent Systems, Stuttgart, Germany, **3** Department of Mechanical Engineering, Shanghai Jiao Tong University, Shanghai, China, **4** Faculty of Kinesiology, University of Calgary, Calgary, Alberta, Canada

* ehalilaj@andrew.cmu.edu

**Data Availability Statement:** All the data, code, and trained models used to produce the results

## Abstract

Knee osteoarthritis is a progressive disease mediated by high joint loads. Foot progression angle modifications that reduce the knee adduction moment (KAM), a surrogate of knee loading, have demonstrated efficacy in alleviating pain and improving function. Although changes to the foot progression angle are overall beneficial, KAM reductions are not consistent across patients. Moreover, customized interventions are time-consuming and require instrumentation not commonly available in the clinic. We present a regression model that uses minimal clinical data—a set of six features easily obtained in the clinic—to predict the extent of first peak KAM reduction after toe-in gait retraining. For such a model to generalize, the training data must be large and variable. Given the lack of large public datasets that contain different gaits for the same patient, we generated this dataset synthetically. Insights learned from a ground-truth dataset with both baseline and toe-in gait trials (N = 12) enabled the creation of a large (N = 138) synthetic dataset for training the predictive model. On a test set of data collected by a separate research group (N = 15), the first peak KAM reduction was predicted with a mean absolute error of 0.134% body weight * height (%BW*HT). This error is smaller than the standard deviation of the first peak KAM during baseline walking averaged across test subjects (0.306%BW*HT). This work demonstrates the feasibility of training predictive models with synthetic data and provides clinicians with a new tool to predict the outcome of patient-specific gait retraining without requiring gait lab instrumentation.

## Author summary

Gait retraining is a conservative intervention for knee osteoarthritis shown to reduce pain and improve function. Although customizing a treatment plan for each patient results in a better therapeutic response, customization cannot yet be performed outside of the gait laboratory, preventing research advances from becoming part of clinical practice. Our work aimed to build a model that accurately predicts whether a patient with knee osteoarthritis will benefit from non-invasive gait retraining using measures that can be easily collected in the clinic. To overcome the lack of large datasets required to train predictive

presented in this manuscript are publicly available at https://simtk.org/projects/predict-kam.

**Funding:** NR is supported by the United States National Science Foundation Graduate Research Fellowship Program under grant numbers DGE1745016 and DGE2140739 (https://www.nsfgrfp.org/). The funder had no role in study design, data collection and analysis, decision to publish, or preparation of the manuscript. Any opinions, findings, and conclusions or recommendations expressed in this material are those of the author(s) and do not necessarily reflect the views of the National Science Foundation.

**Competing interests:** The authors have declared that no competing interests exist.

models, we generated data synthetically (N = 138) based on limited ground-truth examples, and we provide experimental evidence for the model's ability to generalize to real data (N = 15). Our results contribute toward a future in which clinicians can use data collected in the clinic to easily identify patients who would respond to therapeutic gait retraining.

## 1. Introduction

Globally, one in five individuals aged 40 and older are afflicted by knee osteoarthritis, a painful joint disease that still lacks a cure or disease-modifying intervention [1]. Pain is managed pharmaceutically, while structurally cartilage is left to degrade until joint failure, at which point joint replacement surgery is recommended. Although the etiology of the disease is multifactorial, disease progression is known to be exacerbated by high joint loading during ambulation [2]. Osteoarthritis presents more often in the medial compartment than the lateral compartment of the knee partially because the common varus (bow-leg) alignment increases medial knee contact force [3,4]. However, since the forces that stress the tibiofemoral contact surface generally cannot be measured *in vivo*, the knee adduction moment (KAM) is often used as a surrogate of medial compartment knee loading [5]. Both a higher peak KAM [6] and higher KAM impulse [7] are associated with osteoarthritis progression: reducing either or both peaks of the typically two-peaked KAM has therefore been a primary target of non-invasive gait retraining interventions [8].

Biomechanists continue to direct efforts toward conservative interventions that can preserve native joint health to the greatest extent possible [9]. Gait modification strategies to reduce KAM have included decreasing walking speed [10,11], increasing trunk sway [10,12–15], and changing the foot progression angle (FPA) by walking with the toes pointed inward or outward [16–22]. For some, the decrease in walking speed required to reduce peak KAM can be prohibitive to daily living [11]. Increasing trunk sway has been reported to induce back pain and imbalance [14]. In contrast, changing the foot progression angle reduces KAM successfully [23] and is generally preferred by patients, who report minimal discomfort [18]. Walking with the toes pointed inward can reduce the first peak of KAM by lateralizing the foot center of pressure and medializing the knee joint center (Fig 1). After six weeks of retraining with a toe-in gait, knee osteoarthritis patients reduced the first peak of KAM by 0.44% body weight * height (BW*HT) on average, reporting reduced knee pain and improved function at a one-month follow-up [20]. This reduction was comparable to high tibial osteotomy [24] but without the risks associated with the surgery [25].

Customizing the change in foot progression angle to the individual results in a larger peak KAM reduction than a non-custom intervention [17]. However, determining a target foot progression angle is a time-consuming process of acclimation and evaluation that can be performed only in the gait lab. At present, researchers must iteratively evaluate KAM reduction at a range of foot progression angles, relying on force plates and motion capture to compute and compare joint moments. Although proposed methods for estimating KAM using wearable sensors [26,27] or synthetic video data [28] could help reduce reliance on gait lab equipment, these approaches still require gait data collection. No methods currently exist to automate the procedure of predicting KAM reduction in response to gait retraining interventions.

To translate prescription of gait retraining to the clinic, we sought to build a predictive model for KAM reduction with toe-in gait using only features that can be obtained in the clinic without gait analysis tools (Fig 2). For such a model to be generalizable to new patients, large

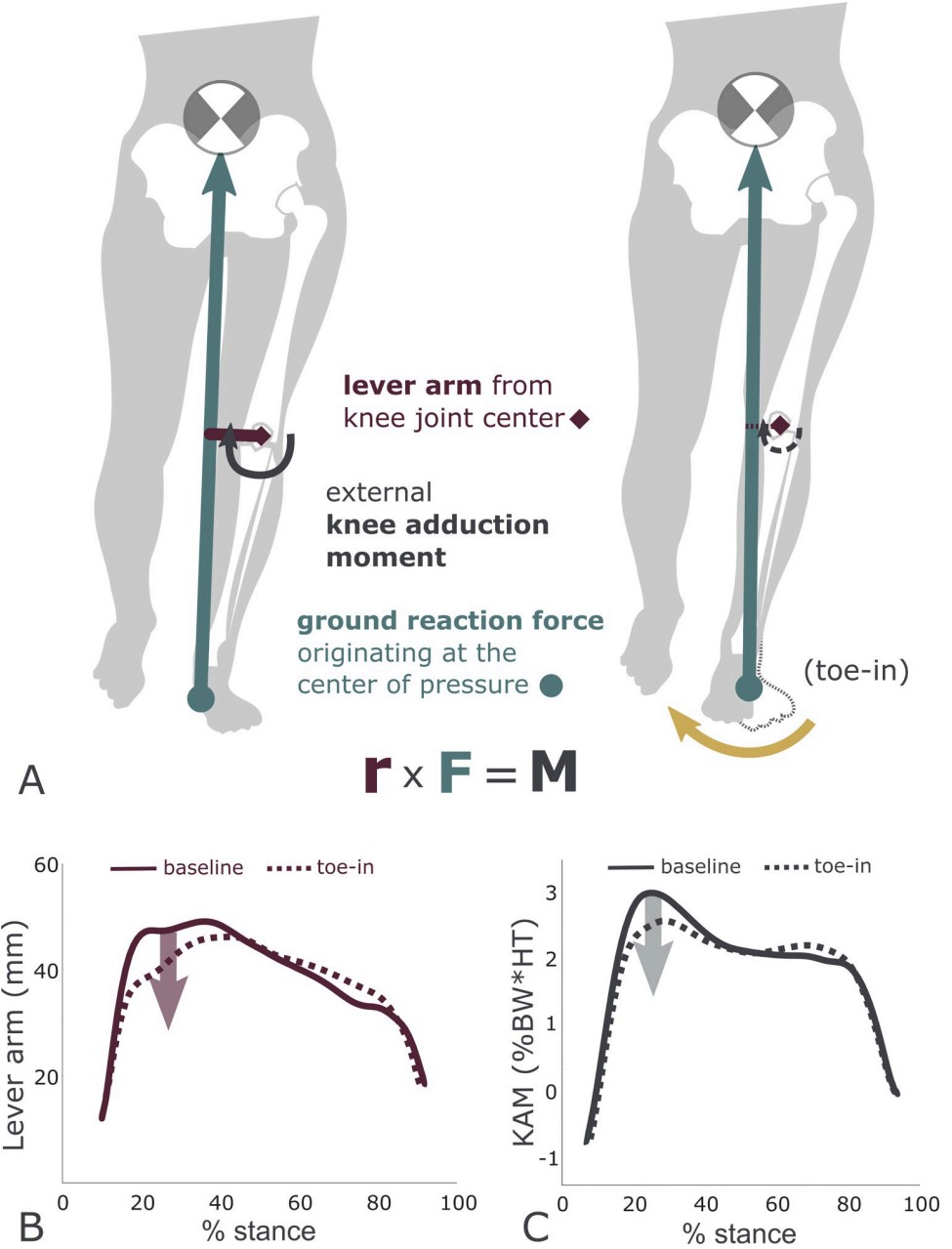

**Fig 1. Toe-in gait reduces the knee adduction moment.** (A) The external moment about the knee is computed using the ground reaction force and lever arm from the knee joint center. Toe-in gait shifts the knee joint center medially and the foot center of pressure laterally in the first half of stance, reducing the KAM. Group average data [18] illustrate how (B) toe-in gait shortens the lever arm and (C) reduces the KAM. Ground reaction force magnitude (not shown in the figure) does not change.

training data with both baseline and toe-in gait are needed, but this specialized dataset does not yet exist. To address this need, we synthesized toe-in gait from baseline walking data from 138 participants. The synthetic toe-in gait was generated using learned gait modification patterns extracted from a ground-truth dataset of 12 participants walking at baseline and toe-in. We evaluated both the synthetic data generation approach and the predictive model on an independent dataset collected in a different laboratory.

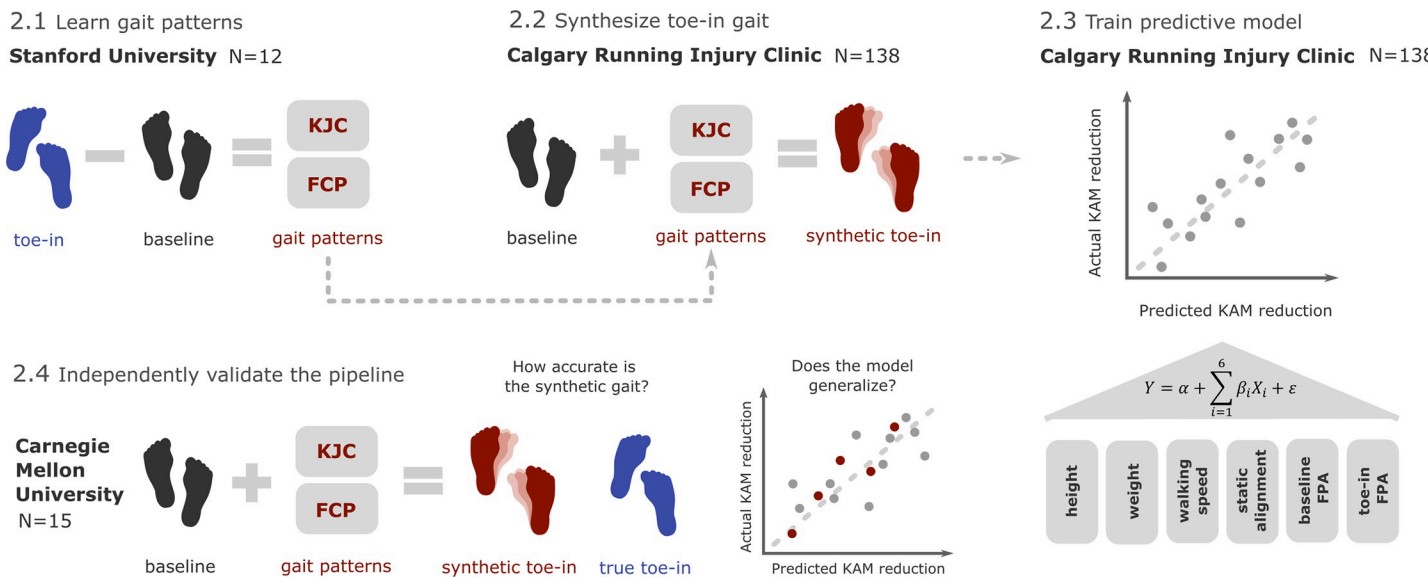

**Fig 2. Study overview.** This study focused on building a regression model that uses minimal clinical data to predict the extent of first peak KAM reduction after toe-in gait retraining. Given the lack of large datasets that contain both baseline and toe-in gaits for the same patient, we generated this dataset synthetically. Gait patterns of knee joint center (KJC) and foot center of pressure (FCP) learned from a ground-truth dataset (N = 12) with both baseline and toe-in gait trials (2.1, Stanford University dataset) enabled the creation of extensive synthetic toe-in data (N = 138) from a dataset that contained only baseline gait trials (2.2, Calgary Running Injury Clinic dataset). A regression model using height, weight, walking speed, limb alignment, baseline FPA, and toe-in FPA was then built to predict KAM reduction (2.3). Both the synthetic data generation approach and the predictive model were tested using data (N = 15) collected by a separate research group (2.4, Carnegie Mellon University dataset).

## 2. Methods

### Ethics statement

All experimental procedures involving human participants were approved by each institution's ethics review boards: the Stanford University Institutional Review Board, the University of Calgary Conjoint Health Research Ethics Board, and the Carnegie Mellon University Institutional Review Board. All participants provided their written informed consent to participate.

### 2.1 Learning gait patterns

Optical motion capture and force plate data from 12 subjects with knee osteoarthritis (Table 1, Stanford University) were used to learn how gait changes with increasing toe-in angle. Subjects walked at a self-selected speed on a split-belt instrumented treadmill under two conditions: walking normally (baseline gait) and walking with toes pointed in (toe-in gait). The last ten steps of the osteoarthritic leg in each condition were used for analysis. We filtered force data at

**Table 1. Summary demographics for participants included in each dataset.**

|  | Stanford University | Calgary Running Injury Clinic | Carnegie Mellon University |
|---|---|---|---|
| *Number of subjects* | 12 | 138 | 15 |
| *Sex* | 7 M / 5 F | 57 M / 81 F | 9 M / 6 F |
| *Height [1]* | 170.7 cm (±8.4 cm) | 171.6 cm (±8.9 cm) | 173.9 cm (±9.3 cm) |
| *Weight [1]* | 77.7 kg (±18.0 kg) | 70.2 kg (±12.4 kg) | 68.0 kg (±11.1 kg) |

[1] Mean (±STD)

15 Hz using a fourth-order zero-lag Butterworth low-pass filter. Foot progression angle was defined in the laboratory horizontal plane as the angle between the anterior-posterior axis and the line connecting the markers placed on the calcaneus and the second metatarsal head. A toe-in angle was defined with respect to a participant's average baseline foot progression angle. A specific toe-in angle was not enforced: mean and standard deviation (± STD) baseline and toe-in angles were 3.97˚ (± 4.91˚) and -5.65˚ (± 4.10˚) respectively.

First, at each step, the foot center of pressure and knee joint center in the mediolateral and anterior-posterior directions were expressed with respect to the pelvis center. The pelvis center was calculated as the centroid of the markers placed on the left and right anterior and posterior superior iliac crests. We then normalized each step from heel strike to toe-off as 0 to 100% stance. For each subject, we represented the foot center of pressure and knee joint center trajectories during *toe-in gait* with respect to that subject's average foot center of pressure and knee joint center trajectories during *baseline gait*. We labeled all 120 toe-in trajectories (10 steps x 12 subjects) by the toe-in angle at that step and grouped them into 1˚ bins from 1˚ to 10˚. The number of steps in each bin ranged from 6 in the 10˚ bin to 21 in the 5˚ bin. Trajectories were averaged within each bin. We represented each of the averaged trajectories using basis splines of order 12 to smooth the prediction. This order was selected from a range of 3 to 20 via cross-validation; increasing beyond 12 yielded no further reduction in root mean squared error (RMSE) between the trajectory and the spline. We then used these gait patterns to synthesize toe-in gait in new subjects by offsetting the learned foot center of pressure and knee joint center trajectory modifications for a given toe-in angle from the new subject's baseline trajectories (Fig 3).

To test whether the learned gait patterns were generalizable across subjects, we carried out exhaustive leave-one-out cross-validation by learning foot center of pressure and knee joint center trajectories from 11 subjects and testing on the left-out subject. We combined the synthesized foot center of pressure and knee joint center trajectories with the subject's baseline ground reaction force, which does not change with toe-in gait, to compute the KAM using the lever-arm method. We compared the synthetic KAM to the subject's ground-truth toe-in KAM using the average RMSE (±STD) for the knee joint center, foot center of pressure, and KAM trajectories, and the mean absolute error (MAE) (±STD) for the first peak KAM across all subjects.

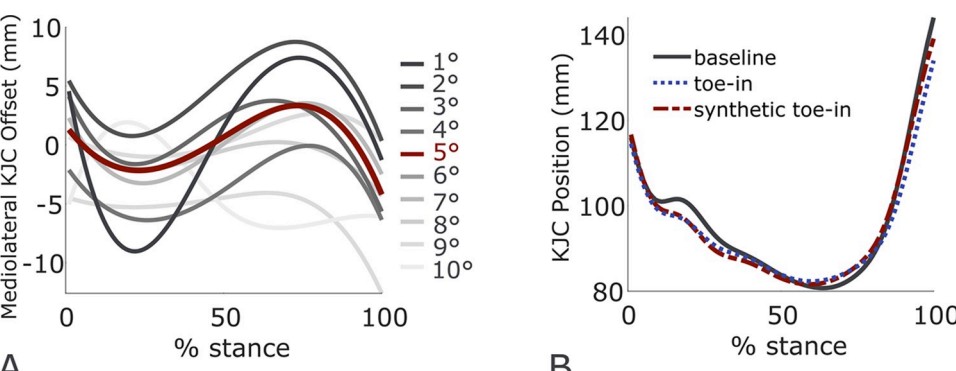

**Fig 3. Synthetic gait generated via learned patterns.** At each toe-in angle from 1˚ to 10˚, all subject trajectories for the foot center of pressure and knee joint center (KJC) were binned, averaged, and fit with a spline. (A) At a given toe-in angle, the trajectory represents the positional offset from baseline gait. (B) Here, knee joint center position in the mediolateral direction was predicted for a representative subject with a 5˚ toe-in angle by adding the learned offset to the baseline trajectory.

## 2.2 Synthesizing toe-in gait data

The learned gait patterns extracted from the ground-truth dataset were used to synthesize toe-in gait from an optical motion capture dataset of 138 subjects performing only baseline gait trials (Table 1, Calgary Running Injury Clinic). All subjects were pain-free at the time of data collection, although some were experiencing a lower extremity running-related injury. Subjects walked at a self-selected speed on an instrumented split-belt treadmill, and data were collected for approximately 2 minutes. We filtered ground reaction force data at 15 Hz using a fourth-order zero-lag Butterworth low-pass filter and removed steps that did not land cleanly on the treadmill's two force plates. We applied the toe-in gait patterns to the average and step-normalized knee joint center and foot center of pressure trajectories for the left leg of each subject, at each toe-in angle from 1˚ to 10˚, resulting in a final dataset of 1380 entries (138 subjects x 10 toe-in angles). The reduction in the first peak KAM from baseline to synthetic toe-in gait was expressed in %BW*HT and used as the response variable of the predictive model.

## 2.3 Training a predictive model

We used the synthetic toe-in gait data to train a predictive model for KAM reduction. To ensure that this model would be useful to clinicians without access to gait lab instrumentation, we used input features that are routinely collected in the clinic (i.e., height, weight, baseline walking speed, static knee alignment) or can be computed with foot-mounted inertial sensors (i.e., baseline foot progression angle and target toe-in angle). The static knee alignment was taken when the subject was standing still and was defined as the angle between the vectors connecting 1) the lateral malleolus and lateral epicondyle and 2) the lateral epicondyle and hip joint center, in the frontal plane, with valgus angle defined as positive. These landmarks can be identified in the clinic by manual inspection and goniometric measurement and may be more accurately estimated in the future with video-based motion capture.

To train a linear regression model to predict first peak KAM reduction, we split data into 80% training, 10% validation, and 10% test sets. Gait cycles from each subject were included in only one set to reduce the risk of inflated performance. All input features were standardized to have zero mean and unit variance using the training data. To quantify model accuracy on synthetic data, we computed the $R^2$ value and the MAE (±STD) between the synthetic first peak KAM reduction and the output of the predictive model. We also computed the mean signed error between the synthetic reduction and model prediction across all toe-in angles for each subject.

## 2.4 Validating gait patterns and predictive model

To evaluate how well the synthetic data generation approach and predictive model generalize to data collected in different settings, we tested the learned toe-in gait patterns and the resulting predictive model using optical motion capture data (Optitrack, Corvallis, USA) collected in a different laboratory (Table 1, Carnegie Mellon University). Fifteen healthy participants walked at baseline and with toe-in gait on an instrumented treadmill (Bertec, Worthington, USA) at a self-selected speed for one minute during both conditions. Participants were guided to maintain a toe-in angle of approximately 5˚ relative to baseline with the use of a commercial device that provides vibration feedback on the shank (SageMotion, Kalispell, USA), but toe-in angles ranged from 3˚ to 10˚. Mean (±STD) angles were 6.46˚ (±5.43˚) for baseline and -6.60˚ (±1.72˚) for relative toe-in, respectively. Synthetic toe-in foot center of pressure, knee joint center and KAM trajectories were generated for each subject, using the gait patterns learned from the Stanford University dataset, as described in Sections 2.1 and 2.2.

The predictive model, described in Section 2.3, was also evaluated on the 15 new subjects using their height, weight, baseline walking speed, static knee alignment, baseline foot

progression angle, and average toe-in angle as input features. We computed the $R^2$ value and the MAE (±STD) between the actual first peak KAM reduction and the predicted KAM reduction, as well as the mean signed error between real and predicted peak KAM reduction. To assess if there was a reduction of accuracy when testing the model on new data, we compared the mean signed errors of the synthetic training data, synthetic test data, and Carnegie Mellon University test data. After testing for normality using the Kolmogorov-Smirnov test, we compared the mean signed errors with a one-way analysis of variance (ANOVA) test and post-hoc Tukey's Honestly Significant Difference (HSD) test for multiple comparisons. We used a significance level of 0.05 for all statistical tests.

## 3. Results

### 3.1 Learned gait patterns

Synthetic toe-in KAM correctly captured that all subjects reduced their first KAM peak. The mean [95% CI] of the actual first KAM peak was 3.065 [2.15, 3.98] %BW*HT during baseline gait and 2.622 [1.71, 3.53] %BW*HT during toe-in gait. The mean [95% CI] predicted first KAM peak was 2.681 [1.76, 3.61] %BW*HT (Fig 4). Predicted knee joint center and foot center of pressure trajectories were more accurate in the mediolateral than in the anterior-posterior direction. The resulting synthetic KAM trajectory was estimated with an RMSE (±STD) of 0.253 (±0.112) %BW*HT (Table 2).

### 3.2 Training and evaluating the predictive model

The predictive model estimated the first peak reduction with an MAE (±STD) of 0.095 (±0.072) %BW*HT (Fig 5) and an $R^2$ of 0.87. The average signed error across all toe-in angles for each of the 108 subjects in the training set was distributed around zero with a 95% CI of

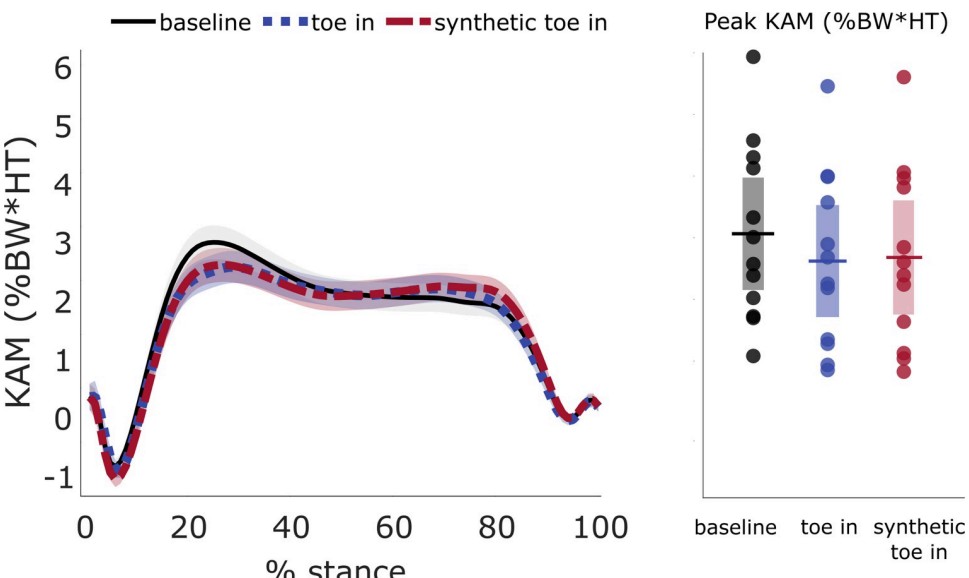

**Fig 4. Validation of toe-in KAM trajectories and first peak KAM reductions (Stanford University dataset).** With leave-one-out cross-validation, the synthetic toe-in KAM trajectory (red, dashed line) from the Stanford University dataset closely matched the ground-truth toe-in KAM (blue, dotted line). Synthetic KAM captured the within-subject reduction in the first peak of KAM relative to baseline (black, solid line) with an MAE of 0.174%BW*HT. The left plot captures mean (±STD) KAM trajectories across subjects, while the right plot shows individual and mean peak KAM with 95% confidence intervals.

**Table 2. Evaluation of the learned gait patterns against ground truth measurements in the Stanford University dataset.**

|  | RMSE (±STD) |
| --- | --- |
| Knee Joint Center (Anterior-Posterior) | 12.7 (±7.8) mm |
| Knee Joint Center (Mediolateral) | 5.6 (±2.4) mm |
| Center of Pressure (Anterior-Posterior) | 13.4 (±4.8) mm |
| Center of Pressure (Mediolateral) | 8.1 (±5.4) mm |
| KAM Trajectory | 0.253 (±0.112) %BW*HT |
|  | MAE (±STD) |
| First KAM Peak | 0.174 (±0.135) %BW*HT |

[-0.020, 0.020]. The mean signed error for the 15 subjects in the synthetic test set was also distributed around zero with a 95% CI of [-0.057, 0.061]. The toe-in angle was the strongest predictor, with a linear weighting coefficient of $\beta_1 = 0.30$ ($p < 0.0001$). Increased valgus angle during static alignment ($\beta_2 = -0.015$, $p = 0.0002$) and increased weight ($\beta_3 = -0.014$, $p = 0.0025$) contributed to a smaller KAM reduction. Baseline foot progression angle ($\beta_4 = -0.0049$, $p = 0.198$), height ($\beta_5 = -0.0041$, $p = 0.398$), and walking speed ($\beta_6 = 0.0034$, $p = 0.375$) did not significantly contribute to a reduction in KAM. When using only the toe-in angle as a feature, KAM reduction was estimated with an MAE (±STD) of 0.096 (±0.073) %BW*HT.

## 3.3 Validating the learned gait patterns and predictive model on a separate dataset

The synthetic toe-in KAM correctly captured that all subjects reduced the first KAM peak. The mean [95% CI] of the actual first peak KAM was 2.602 [2.08, 3.12] %BW*HT during baseline gait and 1.982 [1.47, 2.50] %BW*HT during toe-in. The mean [95% CI] of the predicted first peak of the KAM was 2.006 [1.46, 2.56] %BW*HT (Fig 6). The within-subject variation of the

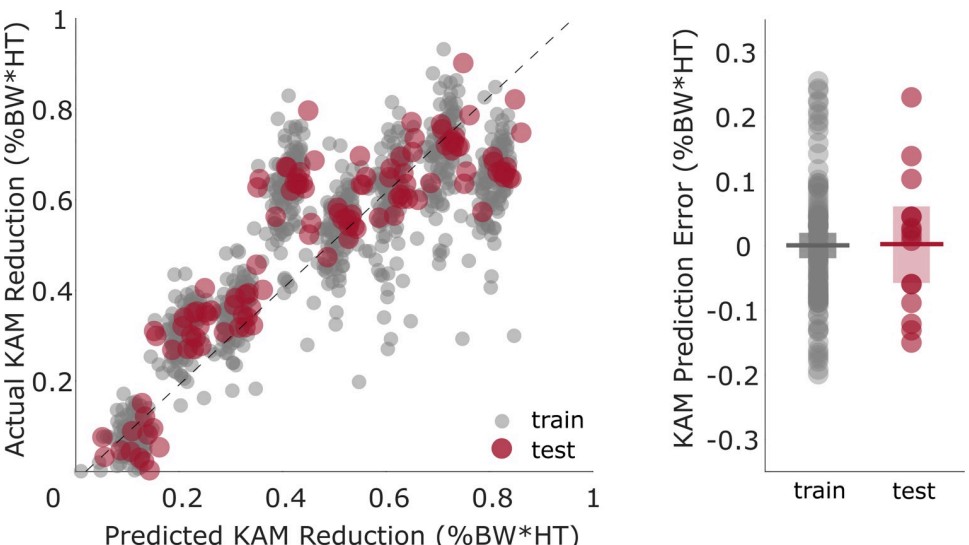

**Fig 5. Prediction of KAM reduction using synthetic training data (Calgary Running Injury Clinic dataset).** Synthetic toe-in data from 108 subjects were used to train the predictive model, which achieved an MAE of 0.0826 (± 0.0628) %BW*HT on the test set of 15 subjects. The line of best fit (dashed line) has an $R^2$ of 0.87. The signed errors (the difference between actual and predicted KAM reduction) of the training and test sets were similarly distributed around zero.

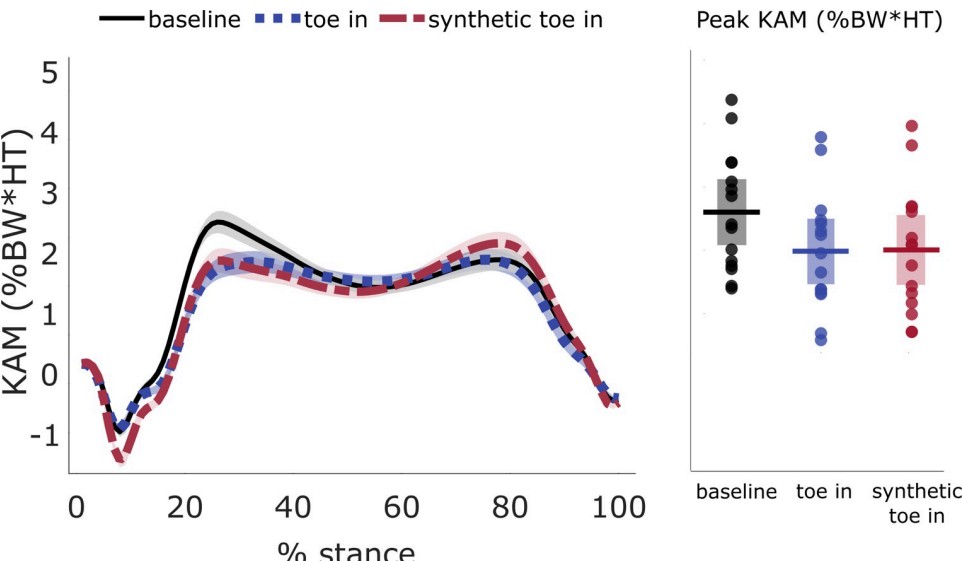

**Fig 6. Validation of synthetic toe-in KAM (Carnegie Mellon University dataset).** The synthetic toe-in KAM trajectory (red, dashed line) closely matched the real toe-in KAM (blue, dotted line). Synthetic KAM captured the within-subject reduction in the first peak of KAM, relative to baseline (black, solid line) with an MAE of 0.170% BW*HT. The left plot captures mean (±STD) KAM trajectories across subjects, while the right plot shows individual and mean peak KAM with 95% confidence intervals.

first KAM peak was greater than the error in the predicted KAM peak: the mean STD of the first KAM peak during baseline walking averaged across all subjects was 0.306%BW*HT. The knee joint center and foot center of pressure for toe-in gait were predicted with similar accuracy to the Stanford University dataset. The predicted knee joint center and foot center of pressure trajectories were again more accurate in the mediolateral direction than in anterior-posterior, and the resulting synthetic KAM trajectory was estimated with an RMSE (±STD) of 0.335 (±0.121) %BW*HT (Table 3).

The predictive model trained on the synthetic data could estimate first peak KAM reduction of the Carnegie Mellon University dataset with an MAE (±STD) of 0.134 (±0.0932) % BW*HT (Fig 7) and an $R^2$ value of 0.55. When using only the toe-in angle, the strongest predictor, as a feature, KAM reduction was estimated with an MAE of 0.187%BW*HT (±0.151% BW*HT). Holding all other inputs constant, increasing valgus angle by 12° or weight by 40 kg resulted in mean peak KAM reduction of less than 0.50%BW*HT. The mean [95% CI] signed error between the predicted and actual first peak KAM reduction was 0.068%BW*HT [-0.017, 0.15]. The mean signed error of the KAM prediction on independently collected data was not

**Table 3. Validation of the learned gait patterns using the Carnegie Mellon University dataset.**

|  | RMSE (±STD) |
|---|---|
| Knee Joint Center (Anterior-Posterior) | 13.3 (±8.2) mm |
| Knee Joint Center (Mediolateral) | 4.5 (±2.7) mm |
| Center of Pressure (Anterior-Posterior) | 15.4 (±8.8) mm |
| Center of Pressure (Mediolateral) | 9.0 (±6.8) mm |
| KAM Trajectory | 0.335 (±0.121) %BW*HT |
|  | MAE (±STD) |
| First KAM Peak | 0.170 (±0.101) %BW*HT |

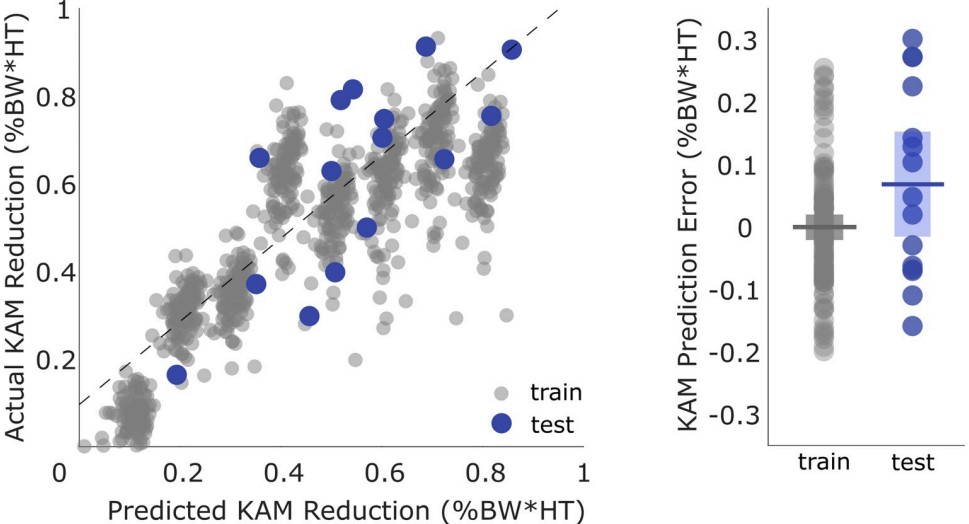

**Fig 7. Validation of the predictive model (Carnegie Mellon dataset).** An independent dataset of toe-in gait from 15 subjects was used to evaluate the predictive model, which achieved an MAE of 0.134%BW*HT (±0.0932%BW*HT). The line of best fit (dashed line) has an $R^2$ of 0.55. The mean signed error of the model was not significantly different between the Carnegie Mellon University test data and the synthetic training or synthetic testing data.

statistically different from the error of the model on training data ($p = 0.082$). We found no significant difference between the mean signed error of the synthetic training and synthetic test data ($p = 0.998$), training and actual test data ($p = 0.0635$), or synthetic test data and actual test data ($p = 0.224$).

## 4. Discussion

The aim of this work was to build a predictive model that uses minimal clinical data to estimate reduction in the first peak of KAM during a toe-in gait retraining. On data collected in a separate gait laboratory, the model could predict KAM reduction with an error of 0.134%BW*HT, which falls within the range of step-to-step KAM peak variation (0.306%BW*HT). This validation on independent data, in addition to the traditional approach of validating on left-out test data, gives us cautious optimism in the model's generalizability. Gait retraining is not yet standard in clinical practice, in part because a full gait analysis is required to identify whether a patient would respond to the therapy. With this predictive model and emerging lightweight haptic feedback systems [29], we hope that clinicians may one day be able to identify responders and prescribe therapeutic gait retraining in a matter of minutes.

Several study characteristics and limitations should be considered when interpreting the reported findings. In this study, KAM was computed using the lever-arm method. Although the inverse dynamics link-segment method is sometimes preferred when real-time KAM estimates are not imperative [8], the two methods show agreement in assessing percent change with a gait intervention, with a mean (±STD) difference of 5% (±14.1%) [30]. When synthesizing toe-in KAM, a 5% error in estimating toe-in peak KAM reduction would have resulted in a difference of 0.0298%BW*HT, which is an order of magnitude less than the Carnegie Mellon University subjects' step-to-step KAM variation (0.306%BW*HT). Another limitation is that some subjects in the Stanford University dataset increased the second peak of the KAM with toe-in, resulting in a falsely predicted second peak increase for the Carnegie Mellon University dataset. However, the average estimated second peak increase was smaller than the estimated first peak reduction, resulting in an overall smaller KAM impulse, which may still provide

therapeutic benefit [31]. *Toe-out* gait modifications typically aim to reduce the second KAM peak [19], so future addition of toe-out gait data may facilitate more accurate prediction of first and second peak changes. Finally, as all of the subjects in the Carnegie Mellon University dataset reduced their first KAM peak with toe-in, it is not clear if the model could accurately predict a clinical non-responder [32,33]. While within the range of reported values [15,19,34], the Carnegie Mellon University dataset average reduction in first peak KAM (0.62 ± 0.226% BW*HT) trended toward being greater than that of the Stanford University dataset (0.44 ± 0.242%BW*HT). This difference may be due to disparities in pain-free mobility between healthy and osteoarthritic cohorts, as well as the method of toe-in gait retraining: here, we provided vibration feedback based on the toe-in angle, whereas the Stanford University researchers provided feedback on the tibial angle in the frontal plane, indirectly inducing a toe-in gait that reduces KAM. The predictive model, built upon patterns learned from a cohort with a smaller KAM reduction, slightly underestimated the real KAM reduction (mean signed error = 0.068%BW*HT), although we did not find this difference to be significant ($p$ = 0.082). Incorporation of several representative datasets from multiple laboratories would further the generalizability of such a model.

KAM reduction with toe-in gait could be predicted using only a limited set of features. As previously reported [15], the toe-in angle positively correlated with KAM reduction. However, using *only* toe-in angle as a predictor estimated KAM reduction to an MAE (±STD) of 0.187 (± 0.151) %BW*HT, compared to 0.134 (±0.0932) %BW*HT using the full set of six features. The next most salient predictor was the increased valgus angle during static alignment, which was related to a smaller KAM reduction. This result supports previous findings that KAM reduction is greater in participants with more varus alignment [34], which may be due to their inherently larger KAM, permitting a greater reduction [35]. Holding all other inputs constant, the model's sensitivity to a 12˚ change toward a more valgus alignment changed the mean KAM peak reduction to less than 0.50%BW*HT, which has previously been used as the clinically meaningful threshold related to risk of disease progression [6,11,28]. The amplitude of this reduction is less than the 16˚ valgus change found to persist six months after a high tibial osteotomy surgery [24], suggesting that the model's sensitivity is physiologically feasible and that static alignment is a relevant predictor for KAM reduction. Although KAM was normalized to height and body weight, higher weight was predictive of a smaller KAM reduction. Here, the model's sensitivity to an added 40 kg resulted in a mean KAM peak reduction that was below the clinically meaningful threshold; at an average height of 170 cm, this added weight would increase BMI from 21 to 35, which is the threshold for obesity [36]. Since knee osteoarthritis patients with a higher weight have a larger weight-normalized KAM than their lean age-matched osteoarthritic controls [37], further investigation is warranted to determine why individuals at a higher weight may be unable to achieve significant reductions to KAM despite their larger KAM.

In a previous study by Boswell et al. [28], a neural network trained on time-discretized 3D anatomical features from 86 subjects was used to classify whether an individual would increase or reduce their first peak KAM with toe-in or toe-out gait modifications, attaining accuracies of up to 85%. Their models predicted the peak KAM during baseline walking with MAE ranging from 0.37–0.49%BW*HT. The most salient features of this model included those related to the position of the pelvis, knee angle in the frontal plane, and sway of the trunk; in the future, a sensor-fusion approach that combines wearable sensing and vision-based motion capture could make use of these additional pelvis and trunk features to improve our predictive model. However, while our model was capable of correctly predicting a KAM reduction for all subjects in the external dataset, we sought to estimate the *extent* of KAM reduction, rather than *classify* subjects based on whether KAM would decrease or not. Furthermore, estimating KAM

reduction from minimal clinical data, as we have done here, instead of computing KAM from cameras or wearable sensors [26,27], eliminates the need for gait analysis. A fast estimate of the KAM reduction with gait retraining may empower clinicians to weigh the expected benefit of retraining against that of other treatment alternatives.

The accuracy of the knee joint center and foot center of pressure predictions were within the error range of joint center location estimates obtained with optical motion capture. Although divergent experimental protocols prevent a meaningful meta-analysis [38], estimates of the knee joint center position have been found to vary from 14 mm to up to 40 mm due to soft tissue artifacts [39–41]. Compared to gold-standard position tracking with intra-cortical bone pins, optical motion capture suffers from skin movement artifacts and inconsistency in marker placement across subjects, experimenters, and laboratories [42]. The inter- and intra-dataset validation of the learned gait patterns showed comparable accuracy between the Stanford University and Carnegie Mellon University datasets, giving us further confidence in this method of generating the synthetic KAM data. The use of these validated toe-in gait patterns may therefore enable reliable predictions of changes in joint kinetics by any research group, without the need for additional motion capture trials.

Accurate prediction of an expected gait change without baseline kinetic data is a significant step toward moving gait retraining prescriptions to the clinic. Collecting gait data with altered foot progression angles is a time-consuming iterative process: after becoming acquainted with the biofeedback paradigm, subjects must acclimate to each new gait pattern, at which point their kinetics may be evaluated. In the absence of universal guidelines on short- and long-term gait retraining learning rates [43], experimenters allow subjects a minimum of 2 minutes to acclimate to the new foot progression angle, with 10–30 minutes of training to enforce the modified gait strategy [17–20]. With the use of the predictive model, this experimental time may be reduced drastically: height, weight, walking speed, and static knee alignment are already commonly measured in the clinic, whereas wearable sensors and vision-based motion capture can one day allow gait kinematics such as foot progression angle to be standard vitals that comprise a patient's unique gait health profile [44].

Modern data science methods offer biomechanists new ways to creatively solve long-standing scientific and clinical challenges [45]. These methods require large and heterogeneous data to avoid overfitting and to generalize well to unseen subjects [46]. Although sharing normative data-sets is becoming more common [47,48], these datasets do not yet exist for more specialized needs such as gait retraining. Here, we demonstrate one promising method to overcome the present paucity of data by generating synthetic data based on limited ground-truth examples. With continued exploration of synthetic data-generation approaches, we hope to move toward a future in which the therapeutic benefit of a potential treatment can be assessed to determine a custom treatment path for any patient.

In conclusion, this work demonstrates that it is possible to predict the extent to which a patient will benefit from gait retraining therapy using clinically available measures. It also illustrates the feasibility of training predictive models with synthetic data. By further harnessing the growing capabilities of emerging motion capture techniques, including ones that fuse wearable sensing and computer vision, these models should empower clinicians to prescribe gait retraining therapy in the clinic.

## Acknowledgments

The authors would like to thank Owen Pearl, as well as the Stanford University and Calgary Running Injury Clinic researchers, for their assistance with data collection. They would also like to thank the participants for their contribution to this study.

## Author Contributions

**Conceptualization:** Nataliya Rokhmanova, Katherine J. Kuchenbecker, Eni Halilaj.

**Data curation:** Nataliya Rokhmanova, Peter B. Shull, Reed Ferber, Eni Halilaj.

**Formal analysis:** Nataliya Rokhmanova, Eni Halilaj.

**Investigation:** Nataliya Rokhmanova, Katherine J. Kuchenbecker.

**Methodology:** Nataliya Rokhmanova, Eni Halilaj.

**Project administration:** Eni Halilaj.

**Resources:** Peter B. Shull, Reed Ferber, Eni Halilaj.

**Supervision:** Katherine J. Kuchenbecker, Eni Halilaj.

**Validation:** Nataliya Rokhmanova.

**Visualization:** Nataliya Rokhmanova.

**Writing – original draft:** Nataliya Rokhmanova, Eni Halilaj.

**Writing – review & editing:** Nataliya Rokhmanova, Katherine J. Kuchenbecker, Peter B. Shull, Reed Ferber, Eni Halilaj.

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
