## [Decision Letter · Decision Letter 0]

9 Feb 2022

Dear Prof. Halilaj,

Thank you very much for submitting your manuscript "Predicting Knee Adduction Moment Response to Gait Retraining with Minimal Clinical Data" for consideration at PLOS Computational Biology. As with all papers reviewed by the journal, your manuscript was reviewed by members of the editorial board and by several independent reviewers. The reviewers appreciated the attention to an important topic. Based on the reviews, we are likely to accept this manuscript for publication, providing that you modify the manuscript according to the review recommendations.

We know these trying times have created challenges for reviewing processes in many situations.  Please know that we very much appreciate and thank you for your patience in this process.

Sincerely,

Jonathan B. Dingwell, Ph.D.

Guest Editor

PLOS Computational Biology

Daniel Beard

Deputy Editor

PLOS Computational Biology

[LINK]

Reviewer's Responses to Questions

**Comments to the Authors:**

Reviewer #1: This manuscript describes a new approach to estimating the potential effects of gait modification on knee adduction moment (KAM), a surrogate for knee loading and an important measure in the study and treatment of knee osteoarthritis. The KAM is typically measured using a sophisticated experimental setup in research gait labs, precluding its widespread assessment in clinical gait labs. In this study, the authors use a limited training dataset, and augment it with a synthesized dataset (based on systematic variations applied to the training dataset), to develop a generalizable statistical model that can estimate how KAM can be modified with changes to gait. They show that using just a handful of basic metrics – many already in use in clinical labs – it is possible to accurately predict the magnitude of KAM reduction following gait retraining, in terms of adopting a toe-in foot posture. This result, and the method as a whole, is encouraging, highlighting the potential for greater accessibility to non-invasive clinical intervention measures in future.

The study is technically sound, the methods are appropriate and the results support the conclusions. The manuscript is written in a clean, concise and intelligible fashion, and the number and quality of figures is appropriate. Beyond a few minor suggestions for improvement (below), I think it is in good shape and should be published. I commend the authors on a well planned and executed study.

Specific comments

Abstract: Provides a good overview, but one key piece of information is missing – detail on the model itself. Just an extra sentence or phrase giving the reader some idea of what kind of model it is, what it involves (input, etc.).

Lines 37-38, ‘medial compartment’: is the medial compartment of the knee generally the one to be more problematic in OA? It might be beneficial, for the broad readership of PLOS Comp Biol, to add some extra detail/justification on this point.

Line 47, “Walking with the toes pointed inward”: I recommend moving this phrase up two sentences, immediately after when FPA is first introduced. This should help with clarity for the non-specialist.

Lines 87-88, “The last ten strides…”: Do you mean strides or steps here? Later on you talk about steps. A single stride comprises two steps. Also, this suggests that both right and left legs are considered in the study, which would be worth explicitly clarifying.

Line 130, “landing cleanly on the force plates”: Isn’t this inconsistent with the use of a treadmill? Some extra clarification is required here.

Line 144: I presume that the ‘static alignment’ is measured when the subject is standing still, correct? An extra clarifying phrase would help here.

Fig. 1: Excellent figure. Are panels (B) and (C) stylistic only, or are they based on actual data? If the latter, this should be explained/cited in the caption as appropriate. Also, the description for panel (A) should use “external moment” instead of just “moment”.

Fig. 3: I am curious as to why the trajectory for the 10 degree bin is so markedly different from the trajectories of the other bins. Is this because of a small sample size or outlier at this extreme foot position? And how much does this affect your model? Some discussion of this, in the appropriate part of the text, would be welcomed.

Fig. 5: In the left panel, I recommend indicating that the dashed line is a line of parity, not some fitted regression line. The same goes for Fig. 7 as well.

Reviewer #2: The manuscript reports the results from a data-driven model predicting the reduction of the first peak of knee adduction moment (KAM) in patients affected by osteoarthritis after toe-in gait retraining. The final objective of this work is to determine whether a patient would benefit from the retraining therapy reducing the experimental time needed to determine the target foot progression angle. The algorithm has been trained with synthetically generated data because of the lack of available large datasets required, and six relevant gait features were considered. The findings demonstrated that the algorithm could quantitatively establish the amount of KAM reduction. The efficacy of training with synthetic gait data could greatly assist clinicians in defining the best treatments for patients without using complex instrumentation.

The manuscript is well written, and I really enjoyed reading it. I only have relatively minor comments:

1. The introduction effectively reports the clinical overview, describing the correlation between higher peaks of KAM and osteoarthritis progression, the possible non-invasive therapies by changing the foot progression angle, and the need of automating the procedure of target foot progression angle's evaluation. However, a better overview of previous data-driven studies addressing similar problems in gait biomechanics would be necessary. It would be very interesting to highlight previous contributions with their limitations and clearly state which of these limitations this study aims to overcome. A study is mentioned in line 291 in the Discussion (reference 36), but a more extensive state-of-the-art would better define this manuscript's contribution and novelty. For instance, a recent study used a feedforward neural network to predict ground reaction forces and joint moments (https://doi.org/10.1016/j.medengphy.2020.10.001). Another one used a neural network to predict KAM from anatomical landmarks obtained in 2D video analysis (https://doi.org/10.1016/j.joca.2020.12.017). There are likely other publications about the use of data-driven approaches in biomechanics that would be interesting to compare.

2. The methodology is rigorous and very well explained. A small clarification could be done in the motivation to choose the six features. Why are these features chosen and not others? Is it just because they are easy to measure? How is the choice of these features more advantageous compared to other studies? (see previous links and https://doi.org/10.1016/j.knee.2020.12.006)

3. The Results are detailed and promising, and the Discussion illustrates well the main achievements and the limitations of the study.

3.1 The Results section ranks the six features from the most significant to the lowest (line 201-206). Successively, it has been verified that the toe-in angle alone could predict the KAM reduction with a small error range. Could the authors discuss their recommendation concerning how many and which features they would use, and briefly give a reason?

3.2 Similarly to the Introduction, the Discussion would benefit from the comparison with the state-of-the-art in data-driven approaches, clearly stating how this research advances compared to similar studies.

**Have the authors made all data and (if applicable) computational code underlying the findings in their manuscript fully available?**

Reviewer #1: Yes

Reviewer #2: Yes

PLOS authors have the option to publish the peer review history of their article (what does this mean?). If published, this will include your full peer review and any attached files.

Reviewer #1: **Yes: **P.J. Bishop

Reviewer #2: No

Figure Files:

Data Requirements:

Reproducibility:

References:

---

## [Decision Letter · Decision Letter 1]

23 Apr 2022

Dear Prof. Halilaj,

We are pleased to inform you that your manuscript 'Predicting Knee Adduction Moment Response to Gait Retraining with Minimal Clinical Data' has been provisionally accepted for publication in PLOS Computational Biology.

Best regards,

Jonathan B. Dingwell, Ph.D.

Guest Editor

PLOS Computational Biology

Daniel Beard

Deputy Editor

PLOS Computational Biology

Reviewer's Responses to Questions

**Comments to the Authors:**

Reviewer #1: Thankyou for attending to my previous comments. I look forward to seeing the revised manuscript published.

Reviewer #2: The authors answered in details to all the minor comments. The new version of the manuscript greatly clarify the contribution of the study and I do not have any other observations to the manuscript.

**Have the authors made all data and (if applicable) computational code underlying the findings in their manuscript fully available?**

Reviewer #1: Yes

Reviewer #2: Yes

PLOS authors have the option to publish the peer review history of their article (what does this mean?). If published, this will include your full peer review and any attached files.

Reviewer #1: **Yes: **P.J. Bishop

Reviewer #2: No

---

## [Editor Report · Acceptance letter]

11 May 2022

PCOMPBIOL-D-21-01686R1 

Predicting Knee Adduction Moment Response to Gait Retraining with Minimal Clinical Data

Dear Dr Halilaj,

I am pleased to inform you that your manuscript has been formally accepted for publication in PLOS Computational Biology. Your manuscript is now with our production department and you will be notified of the publication date in due course.

With kind regards,

Livia Horvath
